# Optimal Calcium Propionate Supplementation in Early-Lactation Dairy Cows Improves Milk Yield and Alters Milk Composition

**DOI:** 10.3390/ani15202995

**Published:** 2025-10-16

**Authors:** Fan Zhang, Xiangfang Tang, Benhai Xiong

**Affiliations:** State Key Laboratory of Animal Nutrition and Feeding, Institute of Animal Sciences, Chinese Academy of Agricultural Sciences, Beijing 100193, China; zhangfan07@caas.cn

**Keywords:** ketosis, negative energy balance, mineral, fatty acid

## Abstract

Calcium propionate is a safe and effective feed additive. Early lactating dairy cows often experience negative energy balance (NEB) due to the rapid increase in nutrient demands for milk production. Supplementing with calcium propionate can alleviate the NEB by promoting gluconeogenesis and inhibiting ketone body synthesis. Our previous studies revealed that calcium propionate supplementation enhances milk performance and rumen fermentation. In this study, we further evaluated its effects on milk mineral and fatty acid composition during early lactation, as well as milk productive performance during the peak of lactation.

## 1. Introduction

The rapidly increasing demand for calcium and energy for milk synthesis during early lactation in dairy cows frequently exceeds dietary intake, increasing the risk of hypocalcemia and ketosis [1]. Hypocalcemia (milk fever) primarily arises from the inability to rapidly mobilize skeletal calcium reserves to meet the calcium output in milk. Clinical hypocalcemia, defined by blood calcium levels < 1.4 mmol/L, affects approximately 5% of cows [2], whereas subclinical hypocalcemia (1.4–2.0 mmol/L) impacts approximately 50% of cows [3,4]. Simultaneously, the immense energy requirement for lactation drives a state of negative energy balance (NEB). When the liver’s capacity to metabolize the ensuing flood of mobilized non-esterified fatty acids (NEFAs) is overwhelmed, it leads to the accumulation of ketone bodies, resulting in ketosis [5]. Clinical ketosis affects 2–15% of cows, while subclinical ketosis is exceptionally common, with a prevalence of 40–60% [6]. These metabolic disorders are particularly prevalent in intensive dairy farming systems, where the selection for high milk yield has intensified the metabolic pressure on cows during the periparturient period. Both hypocalcemia and ketosis in dairy cows can negatively impact their productivity, fertility, and overall health [7,8].

Research has shown that dairy cows experiencing fewer health issues during early lactation tend to have higher milk production and more consistent lactation curves [9]. Thus, mitigating NEB and enhancing calcium availability during early lactation can positively influence milk yield in subsequent periods. The milk composition can provide useful information on the health status of dairy cows during early lactation. Research has demonstrated that the concentrations of certain minerals in milk are influenced not only by dietary element concentrations [10] but also by the health status of the cows [11] and the dietary calcium level, which affects the absorption and metabolism of other minerals (particularly P, Mg, Fe, and Zn) [12,13]. Moreover, milk fatty acid profiles serve as noninvasive indicators of the energy and metabolic status of early-lactating cows by reflecting the extent of body fat mobilization [14,15]. This is because fatty acids derived from adipose tissue differ from those synthesized de novo from dietary sources [16].

Calcium propionate is a valuable source of calcium and gluconeogenic precursors [17,18]. Previous studies have shown that calcium propionate can increase milk yield, prevent hypocalcemia, and alleviate NEB in early-lactating cows [18,19]. It may also influence adipose tissue mobilization [20] and affect milk fatty acid profiles. Dietary supplementation with calcium propionate increases dietary calcium levels and improves the health status of dairy cows, which may consequently influence milk mineral concentrations. However, existing studies have primarily focused on the effects of calcium propionate supplementation on milk yield and conventional milk composition in early lactation. The subsequent effects on peak-lactation performance, milk mineral composition, and fatty acid profiles remain unclear.

This study aimed to evaluate the effects of calcium propionate supplementation on productive performance during the peak of lactation and on milk mineral composition and fatty acid profiles during early lactation. We hypothesized that dietary calcium propionate supplementation during early lactation would enhance subsequent lactation performance by improving mineral homeostasis and altering fatty acid metabolism in early lactation, reflecting a reduced reliance on body tissue mobilization.

## 2. Materials and Methods

The animal experiment was conducted from September 2020 to February 2021 at Beijing Sunlon Livestock Development Co., Ltd. (Beijing, China). The research methods and protocols were carried out according to the standards established by the Institute of Animal Sciences of the Chinese Academy of Agricultural Sciences under the protocol of No. IAS2020-93.

### 2.1. Experimental Design and Animal Management

Thirty-two multiparous Holstein dairy cows in early lactation were randomly assigned to the control (CON), low-calcium propionate (LCaP), medium-calcium propionate (MCaP), and high-calcium propionate (HCaP) groups in a randomized block design on the basis of body weight, parity, and previous 305-day milk yield. A priori power analysis was performed with G*Power (version 3.1.9.6) to ensure an appropriate sample size. All dairy cows were subjected to the same feeding management before calving. Calves were separated from their mothers immediately after birth. The initial characteristics of the dairy cows in each group are presented in Table 1. The cows were housed in individual stalls with ad libitum access to feed. Each cow was provided with a diet that allowed 5% to 10% leftover feed before the next feeding. All the cows had free access to food and water. After calving, the dairy cows in the four groups were offered the same total mixed ratio (TMR) as the basal diet, supplemented with different doses of calcium propionate (Jiangsu Runpu Food Technology Co., Ltd., Lianyungang, Jiangsu, China; purity > 99%): 0, 200, 350, and 500 g/d per cow from calving to DIM 35. The dosage of calcium propionate was determined based on previous studies by Martins et al. [21] and Liu et al. [19]. According to the experimental design, each cow was administered calcium propionate three times a day in approximately equal amounts at the time of feeding. The supplement was administered orally via a stainless-steel pellet gun to ensure accurate intake [22]. From 36 to 125 DIM, all cows in the study were fed the same lactation diet as TMR without supplementation of calcium propionate. All cows were milked at 6:00, 14:00, and 22:00 and offered the same basal diet at 7:00, 14:30, and 18:00. The ingredient composition and nutrient levels of the diets during early lactation (0 to 35 DIM) and the peak of lactation (after 35 DIM) are presented in Table 2. The nutrient requirements for dairy cows were formulated according to NRC (2001) recommendations.

### 2.2. Sample Collection

The amount of feed offered and refused for each cow during the peak of lactation was recorded and weighed daily to calculate dry matter intake (DMI). Feed samples were collected weekly for the analysis of chemical composition. The daily milk yield (36–125 DIM) was automatically recorded by an AfiMilk MPC milk meter (Kibbutz Afikim, Afikim, Israel). Two 50 mL milk samples were collected in sterile centrifuge tubes (Corning Incorporated, Corning, NY, USA) from each cow across three consecutive milking times per day at 7, 21, 35, 65, 95, and 125 DIM. The milk samples were mixed at a 4:3:3 volume ratio for the milk samples collected in the morning, noontime, and evening within a single day [23]. Then, the milk samples collected at 65, 95, and 125 DIM were stored at 4 °C with 2-bromo-2-nitropropane-1,3-diol (D & F Control Systems, Inc., Norwood, MA, USA) for the analysis of milk composition, including milk fat, protein, lactose, and somatic cell count (SCC). The milk samples collected at 7, 21, and 35 DIM were stored at −80 °C without preservative for subsequent analyses of mineral composition and fatty acid profiles.

### 2.3. Analytical Procedures

The dietary samples were dried at 55 °C for 48 h and then passed through a 1 mm screen. The diet was analyzed for dry matter (DM), crude protein (CP), ether extract (EE), starch, calcium (Ca), phosphorus (P), and acid detergent fiber (ADF) content according to AOAC [24] methods 934.01, 954.01, 920.39, 996.11, 968.08, 946.06, and 973.18, respectively. Neutral detergent fiber (aNDF) content was analyzed according to Van Soest et al. [25].

The composition of milk fat, protein, lactose, and SCC was analyzed by a Combi Foss 4000 (Hillerød, Denmark) within 48 h. The 4% FCM (fat-corrected milk) yield and ECM (energy-corrected milk) yield were calculated according to 4% FCM = 0.4 × milk yield + 15 × milk fat yield (NRC, 2001), and the ECM (kg/d) = 0.327 × milk yield + 12.95 × milk fat yield + 7.20 × milk protein yield [26]. The Ca, P, Mg, K, Fe, and Zn concentrations in milk were determined using the methods described by Franzoi et al. [27] and Kandhro et al. [28]. Briefly, triplicate 10 g milk samples were weighed into Teflon digestion vessels. Approximately 10.0 mL of a prepared mixture containing nitric acid and perchloric acid at a ratio of 10:1 (*v/v*) was added to the digestion vessel and heated in a Galanz microwave oven (900 W, Galanz Enterprise Group, Foshan, Guangdong, China) until the digestive fluid became colorless, transparent or yellowish. The contents of the vessels were cooled to room temperature and then filtered into a 25 mL volumetric flask by a filter funnel and paper. The volumetric flask was diluted to a volume of 25 mL with ultrapure water for the analysis of milk mineral concentrations. A blank test was also conducted in the same manner without the addition of the milk sample. Inductively coupled plasma atomic emission spectrometry (ICP–OES) with an Agilent 5100 ICP–OES (Santa Clara, CA, USA) was used to determine Ca, P, Mg, K, Fe, and Zn contents at wavelengths of 315.887, 766.491, 239.5, 279.079, 178.287, and 206.2 nm, respectively [29]. Single element solutions (Inorganic Ventures, Christiansburg, VA, USA) were diluted with ultrapure water to prepare calibration solutions at concentrations ranging from 0 to 100 mg/L for Ca, P, Mg, and K and from 0 to 4 mg/L for Fe and Zn [27].

Milk fatty acid profiles were analyzed via an Agilent 8860 gas chromatograph (Agilent Technologies Inc., Santa Clara, CA, USA) equipped with an HP-88 capillary column (100 m × 0.25 mm ID, 0.20 µm film thickness; Agilent, Santa Clara, CA, USA), following the methods of Sun et al. [30]. The milk samples used in the experiments were thawed at 4 °C, after which 2 g of each milk sample was collected into a 10 mL centrifuge tube. A hexane–isopropanol mixture at a ratio of 3:2 (*v/v*) was added to the tube at a volume of 4 mL and mixed for 2 min. Subsequently, 2 mL of 66.7 g/L sodium sulfate solution was added to the tube and vortexed for 2 min. The solution was then centrifuged at 2500× *g* for 10 min. The upper solution was transferred to a high-temperature resistant tube. Then, 2 mL sodium hydroxide methanol solution (2.0 g/100 mL) was added into the solution for saponification of the fat for 30 min at 50 °C. After cooling to room temperature, 2 mL of acetyl chloride in methanol (1:10 *v/v*), prepared from acetyl chloride (Aladdin, Shanghai, China) and methanol (Macklin, Shanghai, China), was added to the mixture, and the mixture was methyl esterified at 90 °C for 2 h. The mixture was then cooled to room temperature, and 2 mL of ultrapure water was added. The mixture was extracted with hexane 3 times, diluted to 10 mL, and 0.5 g of sodium sulfate was added. The supernatant was filtered into vials and used for the analysis of fatty acid profiles gas chromatography. The initial temperature of the gas chromatographic column was 120 °C (10 min), which was subsequently increased to 230 °C at a rate of 1.5 °C/min and maintained at 230 °C for 35 min. The sample injector and detector temperatures were set at 250 °C and 300 °C, respectively. The carrier gas was nitrogen, and the pressure was 190 kPa. The fatty acid methyl ester chromatograms of the milk samples were compared with those of the standard solution (Anpel Laboratory Technologies Inc., Shanghai, China) to identify and quantify the fatty acids. Additionally, the proportions of saturated fatty acid (SFA), unsaturated fatty acid (UFA), monounsaturated fatty acid (MUFA), polyunsaturated fatty acid (PUFA), short-chain fatty acid (SCFA), medium-chain fatty acid (MCFA), long-chain fatty acid (LCFA), and total trans fatty acid contents were calculated on the basis of the carbon chain length and degree of unsaturation of the individual fatty acid concentration [31].

### 2.4. Statistical Analysis

The study employed a randomized block design with repeated measures. The MIXED procedure of SAS 9.4 was used to analyze all the data. The normality of data distribution and homoscedasticity were tested via the UNIVARIATE procedure in SAS. The repeated measures model included the fixed effects of treatment, block, time, the interaction of treatment and time, and the random effects of cows. Sampling time was treated as a repeated measure, and the compound symmetry covariance structure, which yielded the lowest corrected Akaike information criterion [32], was selected for the analysis model. Orthogonal polynomial contrasts in SAS were used to assess the linear and quadratic effects of calcium propionate supplementation levels. Additionally, orthogonal coefficients for the unequally spaced levels of calcium propionate supplementation (0, 200, 350, and 500 g/d) were generated via the IML procedure in SAS. Duncan’s multiple range test was used for multiple comparisons among treatments. Statistical significance was defined as *p* ≤ 0.05, and a trend was defined as 0.05 < *p* ≤ 0.10.

## 3. Results

### 3.1. Productive Performance During Peak Lactation

The results for productive performance during the peak of lactation (36–125 DIM) are shown in Table 3. As calcium propionate supplementation increased, the milk yield tended to increase linearly during the peak of lactation (*p* = 0.09), whereas the content of lactose in milk decreased linearly (*p* = 0.02). However, early-lactation calcium propionate supplementation had no significant effect on DMI, 4% FCM, ECM, milk fat, milk protein, or SCC during the peak of lactation.

### 3.2. Milk Mineral Composition

The effects of calcium propionate supplementation on milk mineral composition during early lactation are presented in Table 4. This study revealed that calcium propionate supplementation did not affect the concentrations of milk Ca or Fe in early-lactation dairy cows. Improving the amounts of calcium propionate supplementation levels quadratically increased the milk K concentrations (*p* < 0.001), with the lowest value observed in the HCaP group, but linearly decreased the milk Mg (*p* = 0.01) and *p* (*p* < 0.01) concentrations. The milk Mg and P concentrations also showed quadratic trends (*p* < 0.10), peaking in the LCaP group. The concentration of Zn in the milk tended to change quadratically with increasing calcium propionate supplementation levels (*p* < 0.10), peaking in the LCaP group. Additionally, the milk K and Fe concentrations were affected by treatment × time interactions (*p* < 0.05) (Figure 1). As shown in Figure 1, the milk K concentration in the HCaP group was significantly lower than that in the other groups at 21 DIM (*p* = 0.002). At 35 DIM, the milk K concentrations in the LCaP and MCaP groups were significantly greater than those in the CON and HCaP groups (*p* < 0.001), with the highest value in the LCaP group. Furthermore, the milk Fe concentration tended to be lower in the calcium propionate-supplemented groups than in the CON group at 21 DIM (*p* < 0.10), but no differences were observed at 7 and 35 DIM.

### 3.3. Milk Fatty Acid Profiles

The effects of calcium propionate supplementation on milk fatty acid profiles in early-lactating dairy cows are shown in Table 5. Increasing the concentration of calcium propionate linearly decreased the proportions of C6:0, C8:0, and C12:0 (*p* < 0.05). Moreover, the proportion of 18:2 *cis*-9, 12 showed both linear and quadratic effects (*p* < 0.01), which were significantly lower in the MCaP and LCaP groups than in the CON group (*p* < 0.01). Increasing the calcium propionate supplementation level quadratically affected the proportions of C17:0 (*p* = 0.01) and C20:4 *cis*-5, 8, 11, and 14 (*p* < 0.001), which were all highest in the MCaP group. The proportions of C15:1 *cis*-10, C18:1 *trans*-9, and C20:2 *cis*-11, 14 tended to initially increase but then decreased as the supplementation level of calcium propionate increased (*p <* 0.10). The proportion of C20:3 *cis*-11,14,17 in milk tended to show a quadratic response in cows supplemented with calcium propionate (*p* < 0.10), with the lowest value observed in the MCaP group. Furthermore, calcium propionate supplementation linearly increased the C10:0 proportion (*p* < 0.05) and tended to reduce the proportion of C18:3 *cis*-9, 12, and 15 (*p* < 0.10). The contents of the other fatty acids did not significantly differ among the four treatments. As shown in Figure 2, the proportions of C16:0, C17:0, C18:2 *cis*-9, 12, and C20:4 *cis*-5, 8, 11, 14 were influenced by the interaction between treatment and time (*p* < 0.05). At 7 DIM, the MCaP group presented the highest proportions of C16:0, C17:0, and C20:4 *cis*-5, 8, 11, and 14 compared with the other groups (*p* < 0.05). Cows supplemented with calcium propionate had lower proportions of C18:2 *cis*-9,12 than did those in the CON group at 7 and 35 DIM (*p* < 0.05).

Descriptive statistics of the milk fatty acid profiles are presented in Table 6. The proportion of PUFA was quadratically affected by calcium propionate supplementation (*p* = 0.01) and was lowest in the MCaP group. Increasing the calcium propionate dose quadratically affected the total trans fatty acid proportion (*p* < 0.10), with the greatest value observed for cows in the MCaP group.

## 4. Discussion

Most dairy cows reach their peak milk yield between 45 and 100 DIM [33], after which the milk yield gradually decreases. In the current study, the higher milk yield in the groups supplemented with calcium propionate than in the CON group suggested that calcium propionate supplementation to dairy cows during early lactation had a positive impact on lactation performance. High lactation persistency, associated with health status in early lactation, is linked to a slow decrease in milk yield after peak production [34]. The peak lactation performance is significantly influenced by nutrition and management practices during early lactation. Our previous study showed that calcium propionate can improve milk yield in dairy cows during early lactation [18]. Furthermore, milk yield during early lactation is positively correlated with extended lactation performance [35]. However, high milk production also increases the risk of elevated SCC [36]. Notably, milk lactose, which serves as the predominant osmoregulatory substance in milk, is negatively associated with SCC in milk [37]. In the current study, all the calcium propionate-supplemented groups had higher milk yields than the CON group. Consequently, the increase in both milk SCC and milk yield may contribute to the linear decrease in milk lactose content. Despite this reduction, the average milk lactose yield in the calcium propionate-supplemented groups was similar to the CON group.

The milk Ca concentration in the study was not affected by dietary calcium propionate feeding levels. The blood Ca concentration is strictly regulated by parathyroid hormone, calcitonin, and 1,25-dihydroxyvitamin D3, which control Ca absorption, excretion, and bone metabolism [38]. The lowest blood calcium levels occur approximately 12 to 24 h after calving [39]. Blood calcium levels are subsequently maintained at a normal level through the mobilization of bone calcium. Accordingly, the finding that blood Ca concentration was unchanged by calcium propionate supplementation at 7, 21, and 35 DIM in our previous study [40] was associated with the unchanged milk Ca concentration. The supplementation of calcium in feed is beneficial for reducing the mobilization of bone calcium. Additionally, the dietary Ca concentration is inversely related to the milk P concentration [41]. It has been reported that diets with high levels of Ca can adversely affect Mg absorption in ruminants [42,43]. Therefore, in this study, the milk Mg and P concentrations during early lactation decreased linearly with increasing levels of calcium propionate.

K plays roles in regulating acid-base balance, maintaining osmotic pressure, transducing signals, transmitting nerve impulses, and contracting muscle [44]. Toscano et al. [45] reported a negative correlation between the milk K concentration and the serum β-hydroxybutyrate (BHB) concentration. Dietary calcium propionate supplementation can decrease the blood BHB concentration [19], which may increase the milk K concentration. Increasing the dietary calcium concentration can increase K absorption [46] and the serum K concentration [47] in dairy cows. However, the milk K concentration decreased when calcium propionate was supplemented at 500 g/d. This increase was accompanied by a decrease in DMI in the HCaP group (21.20 kg/d) compared with the MCaP group (22.87 kg/d) in early lactation, as previously reported [18]. Zinc is important for immune function, cell division, and protein synthesis [48]. The improved metabolic status resulting from dietary supplementation with calcium propionate [20] might enhance Zn absorption in the intestines of dairy cows. However, diets high in Ca can reduce Zn absorption and balance [49]. Consequently, the milk Zn concentration tended to change quadratically with increasing calcium propionate feeding levels during early lactation, peaking in the LCaP group.

Because of the high energy requirement of milk production, dairy cows experience NEB during early lactation. The milk fatty acid profiles of early-lactating dairy cows are useful indicators for identifying NEB status [50]. When a cow is in NEB status, a greater percentage of preformed fatty acids from body fat reserves and a lower percentage of de novo fatty acids from the diet are used to produce milk fat. Propionate plays a crucial role in alleviating NEB in early-lactating dairy cows by promoting glucose synthesis. The levels of short- and medium-chain fatty acids, which are synthesized primarily de novo in the mammary glands of dairy cows, decrease during NEB in early lactation [50]. Although the sum of the SFAs did not differ among the four groups in this study, the proportions of C6:0, C8:0, and C12:0 linearly decreased with increasing levels of calcium propionate. Many studies have demonstrated that dietary supplementation with monensin, an ionophore antibiotic, can alter rumen bacterial population fermentation toward increasing the propionate proportion and decreasing the acetate: propionate ratio, consequently reducing the proportions of short-chain fatty acids in the milk of dairy cows [51,52]. Therefore, in this study, the reduced proportions of these fatty acids may be partly attributed to the increased propionate intake associated with increased levels of calcium propionate. Notably, C17:0 is synthesized de novo from ruminal propionate [53]. In a study by Zhang et al. [18], dairy cows supplemented with calcium propionate at 500 g/d presented decreased DMI. Consequently, increased supplementation with calcium propionate resulted in a quadratic response to the milk C17:0 proportion during early lactation in the current study. The results of Churakov et al. [50] showed that the C18:0 and C18:1 *cis*-9 concentrations in milk were the best variables for detecting the severity of NEB in cows. Pacheco-Pappenheim et al. [54] also reported that dairy cows with NEB status mobilized more C16:0, C18:0, and C18:1 *cis*-9 from body fat reserves, leading to increased proportions of these fatty acids in milk fat. The previous results of Zhang et al. [20] showed that increasing calcium propionate feeding levels quadratically increased milk yield, with the greatest value observed in the MCaP group. However, calcium propionate supplementation did not significantly alter the proportions of these fatty acids in milk in the present study, suggesting that the extra energy requirement for increased milk yield was derived from feed nutrition (22.87 kg/d in the MCaP group vs. 21.71 kg/d in the CON group in early lactation) [18] rather than body fat reserves. Wang et al. [55] reported that inulin could reduce the proportion of C18:2 *cis*-9,12 (linoleic acid) in milk and increase the propionate concentration in the rumen. The decrease in C18:2 *cis*-9, 12 may be related to the increased supplementation of propionate, which improved the biohydrogenation of fatty acids by rumen microorganisms and quadratically changed the toxic effects of PUFAs, as suggested by Lock et al. [56]. Further research should investigate how calcium propionate affects the biohydrogenation of fatty acids in early-lactating dairy cows.

The energy status of dairy cows in early lactation affects the origin of fatty acids for the synthesis of milk fat in the mammary gland [54]. It has been reported that body fat mobilization due to NEB increases the proportion of LCFAs (i.e., C > 16) in milk [16]. However, in this study, calcium propionate did not affect the proportion of LCFAs. It was hypothesized that the increased energy requirement for increased milk yield in the LCaP and MCaP groups [20] was met through dietary intake; therefore, fat mobilization was not affected. PUFAs, which are exclusively obtained from dietary sources [57], were lower in the groups supplemented with calcium propionate than in the CON group. This may be associated with the increased milk production in these groups supplemented with calcium propionate in our previous study [20].

## 5. Conclusions

Supplementing with 350 g/d of calcium propionate during early lactation was identified as an optimal strategy for enhancing peak milk production in dairy cows. This dosage not only improved energy status but also systematically altered milk composition, characterized by increased K concentration and reduced C18:2 *cis*-9,12 proportion.

## Figures and Tables

**Figure 1 animals-15-02995-f001:**
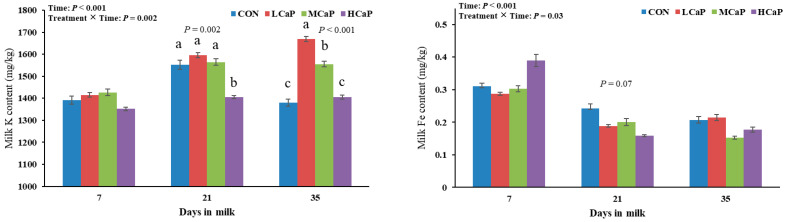
Effects of dietary supplementation with different levels of calcium propionate on milk mineral compositions, including K and Fe, of dairy cows during early lactation (1–35 DIM). CON: without calcium propionate; LCaP, MCaP, and HCaP contained 200, 350, and 500 g/d calcium propionate per cow, respectively. a, b, c Means with no common superscript in the bar graph are significantly different (*p* < 0.05). The error bars represent the standard error. The *p* value at a time point indicates a significant difference among the groups at that time.

**Figure 2 animals-15-02995-f002:**
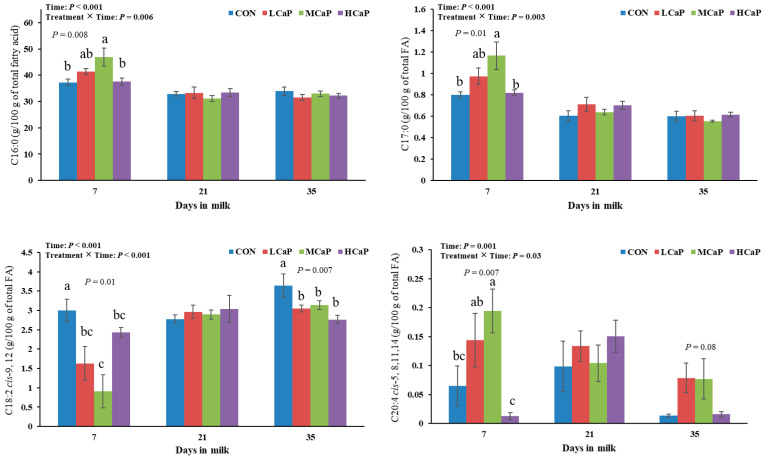
Effects of dietary supplementation with different levels of calcium propionate on milk fatty acid (FA) profiles, including the Fe and K contents, of dairy cows during early lactation (1–35 DIM). CON: without calcium propionate; LCaP, MCaP, and HCaP had 200, 350, and 500 g/d calcium propionate per cow, respectively. a, b, c Means with no common superscript in the bar graph are significantly different (*p* < 0.05). The error bars represent the standard error. The *p* value at a time point indicates a significant difference among the groups at that time.

**Table 1 animals-15-02995-t001:** Initial characteristics of the dairy cows in each group.

Items	Treatments ^1^			
	CON	LCaP	MCaP	HCaP
Body weight (kg)	782 ± 22.6	790 ±26.9	777.8 ± 31.2	768 ± 24.9
Parity	3.33 ± 0.56	3.17 ± 0.31	3.50 ± 0.22	3.17 ± 0.31
Previous 305-day milk yield (kg)	12,710 ± 303	12,720 ± 672	12,522 ± 744	12,741 ± 832

^1^ CON, LCaP, MCaP, and HCaP had 0, 200, 350, and 500 g/d calcium propionate per cow, respectively.

**Table 2 animals-15-02995-t002:** Ingredients and chemical composition of the diets during early lactation and peak lactation.

Items ^1^	1–35 DIM	>35 DIM
Ingredients, % of DM		
Concentrate for early lactation ^2^	41.9	—
Concentrate for peak lactation ^3^	—	37.0
Full-fat extruded soybean	—	0.35
Distiller’s dried grain with soluble	—	0.84
Molasses	—	1.95
Alfalfa silage	—	1.77
Cotton seed	2.28	3.58
Steam-flaked corn	3.58	8.27
Sprouting corn bran	2.19	1.98
Fat powder ^4^	1.14	1.03
Megalac	0.50	0.78
Wet brewers’ grains	3.73	4.53
Pelleted beet pulp	1.31	1.03
Alfalfa hay	9.90	5.89
Oat hay	2.16	2.04
Corn silage	31.3	28.96
Chemical composition, % of DM		
Dry matter (DM), % of fresh weight	50.0	50.5
Crude protein (CP)	17.7	16.4
Net energy for lactation (NE_L_) ^5^, MJ/kg DM	7.20	7.30
Ether extract	4.20	4.80
Neutral detergent fiber (aNDF)	28.0	29.7
Acid detergent fiber (ADF)	15.9	17.8
Starch	23.5	26.0
Ash	9.10	8.90
Ca	0.85	0.87
P	0.42	0.47

^1^ The diet for early lactation was fed to dairy cows from calving to d 35 postpartum; the diet for lactation was fed to dairy cows after d 36 postpartum. ^2^ The concentrate for early lactation was manufactured by Beijing Sanyuan Seed Technology Co., Ltd. (Beijing, China) with the chemical composition of DM, 88.50%; CP, 23.91%; aNDF, 13.20%; ADF, 7.40%; Ash, 13.1%; Ca, 1.41%; P, 0.58%; K, 1.20%; Mg, 0.58%; Na, 0.99%; Cu, 46.25 mg/kg; Fe, 80.30 mg/kg; Zn, 136.8 mg/kg; vitamin A, 20.53 kIU/kg; vitamin D, 3548.5 IU/kg; and vitamin E, 116.9 IU/kg. ^3^ The concentrate for peak lactation was manufactured by Beijing Sanyuan Seed Technology Co., Ltd. (Beijing, China) with the chemical composition of DM, 88.30%; CP, 20.5%; aNDF, 14.40%; ADF, 7.90%; Ash, 12.7%; Ca, 1.37%; P, 0.67%; K, 1.32%; Mg, 0.58%; Na, 0.93%; Cu, 38.98 mg/kg; Fe, 171.3 mg/kg; Zn, 117.9 mg/kg; vitamin A, 16.60 kIU/kg; vitamin D, 2766 IU/kg; and vitamin E, 150.7 IU/kg. ^4^ Fat power (C16:0 > 90%) is produced by Yihai Kerry Foodstuffs Industries (Tianjin) Co., Ltd. (Tianjin, China). ^5^ NE_L_ was calculated according to NRC (2001).

**Table 3 animals-15-02995-t003:** Effects of dietary supplementation with different levels of calcium propionate during early lactation on the productive performance of dairy cows during the peak of lactation (36–125 DIM).

Items	Treatments ^1^				SEM ^2^	*p* Value ^3^	
	CON	LCaP	MCaP	HCaP		L	Q
DMI, kg/d	29.93	30.08	31.71	31.27	0.39	0.11	0.81
Yield							
Milk, kg/d	49.36	50.62	53.67	52.76	0.82	0.09	0.62
4% FCM ^4^, kg/d	49.65	51.15	55.82	53.27	1.16	0.14	0.44
ECM ^5^, kg/d	53.71	55.24	60.11	57.24	1.11	0.13	0.38
Fat, kg/d	1.99	2.06	2.28	2.14	0.06	0.25	0.46
Protein, kg/d	1.63	1.67	1.77	1.70	0.02	0.18	0.33
Lactose, kg/d	2.62	2.64	2.79	2.75	0.04	0.20	0.78
Milk composition							
Fat, %	4.03	4.09	4.27	4.00	0.10	0.91	0.41
Protein, %	3.33	3.32	3.31	3.23	0.03	0.33	0.57
Lactose, %	5.31	5.22	5.19	5.20	0.02	0.02	0.23
SCC ^6^, ×1000 cell/mL	147.8	268.8	199.9	192.0	39.30	0.76	0.43

^1^ CON, LCaP, MCaP, and HCaP had 0, 200, 350, and 500 g/d calcium propionate per cow, respectively. ^2^ SEM: Standard error of the mean. ^3^ L: Linear effects of calcium propionate supplementation; Q: Quadratic effects of calcium propionate supplementation. ^4^ 4% FCM: 4% fat-corrected milk. ^5^ ECM: Energy-corrected milk. ^6^ SCC: Somatic cell count.

**Table 4 animals-15-02995-t004:** Effects of dietary supplementation with different levels of calcium propionate on the milk mineral composition of dairy cows during early lactation (1–35 DIM).

Items, mg/kg	Treatments ^1^				SEM ^2^	*p* Value ^3^	
	CON	LCaP	MCaP	HCaP		L	Q
Ca	1210	1252	1172	1210	15.78	0.59	0.79
P	1009	1027	968.8	917.2	13.30	0.002	0.06
Mg	114.7	120.8	109.3	107.8	1.42	0.01	0.07
K	1441	1560	1514	1387	14.22	0.07	<0.001
Fe	0.25	0.23	0.22	0.24	0.01	0.47	0.15
Zn	4.10	4.66	4.04	4.06	0.10	0.51	0.09

^1^ CON, LCaP, MCaP, and HCaP had 0, 200, 350, and 500 g/d calcium propionate per cow, respectively. ^2^ SEM: Standard error of the mean. ^3^ L: Linear effects of calcium propionate supplementation; Q: Quadratic effects of calcium propionate supplementation.

**Table 5 animals-15-02995-t005:** Effects of dietary supplementation with different levels of calcium propionate on the milk fatty acid proportions (% of total fatty acids) of dairy cows during early lactation (1–35 DIM).

Items	Treatments ^1^				SEM ^2^	*p* Value ^3^	
	CON	LCaP	MCaP	HCaP		L	Q
C4:0	2.24	2.30	2.34	2.32	0.11	0.80	0.87
C6:0	1.50	1.28	1.33	1.18	0.05	0.04	0.75
C8:0	0.90	0.73	0.75	0.70	0.03	0.02	0.31
C10:0	1.53	1.55	1.44	1.96	0.07	0.02	0.31
C11:0	0.061	0.060	0.084	0.052	0.007	0.94	0.29
C12:0	2.19	1.71	1.73	1.66	0.08	0.03	0.23
C13:0	0.088	0.10	0.090	0.076	0.007	0.48	0.23
C14:0	7.71	6.65	6.97	6.72	0.21	0.12	0.32
C14:1 *cis*-9	0.72	0.68	0.70	0.73	0.03	0.87	0.44
C15:0	0.66	0.61	0.62	0.64	0.03	0.76	0.50
C15:1 *cis*-10	0.10	0.14	0.12	0.060	0.01	0.30	0.06
C16:0	34.71	35.40	36.95	34.42	0.65	0.82	0.12
C16:1 *cis*-9	2.02	2.40	2.40	2.23	0.09	0.35	0.11
C17:0	0.67	0.76	0.79	0.71	0.02	0.22	0.01
C17:1 *cis*-10	0.32	0.33	0.28	0.37	0.01	0.45	0.16
C18:0	13.35	12.53	11.64	13.17	0.36	0.59	0.11
C18:1 *cis*-9	25.72	27.77	26.71	28.52	0.89	0.37	0.94
C18:1 *trans*-9	0.70	1.04	1.13	0.84	0.08	0.41	0.06
C18:2 *cis*-9, 12	3.14	2.54	2.31	2.74	0.10	0.02	0.002
C18:2 *trans*-9, 12	0.027	0.032	0.033	0.039	0.003	0.14	0.80
C18:3 *cis*-6, 9, 12	0.067	0.074	0.082	0.080	0.007	0.45	0.70
C18:3 *cis*-9, 12, 15	0.079	0.087	0.11	0.14	0.01	0.08	0.51
C20:0	0.16	0.14	0.20	0.17	0.01	0.45	0.99
C20:1 *cis*-11	0.32	0.30	0.31	0.28	0.02	0.44	0.90
C20:2 *cis*-11, 14	0.044	0.092	0.091	0.065	0.01	0.47	0.08
C20:3 *cis*-8, 11, 14	0.091	0.095	0.090	0.071	0.006	0.09	0.20
C20:3 *cis*-11, 14, 17	0.099	0.087	0.075	0.11	0.007	0.92	0.09
C20:4 *cis*-5, 8, 11, 14	0.058	0.12	0.12	0.066	0.01	0.74	<0.001
C20:5 *cis*-5, 8, 11, 14, 17	0.010	0.016	0.014	0.015	0.002	0.34	0.55
C21:0	0.039	0.067	0.038	0.042	0.009	0.86	0.46
C22:0	0.087	0.095	0.99	0.12	0.01	0.14	0.48
C22:1 *cis*-13	0.014	0.021	0.029	0.024	0.005	0.36	0.27
C22:2 *cis*-13, 16	0.0085	0.022	0.016	0.018	0.003	0.31	0.29
C22:6 *cis*-4, 7, 10, 13, 16, 19	0.045	0.050	0.050	0.047	0.009	0.91	0.82
C23:0	0.022	0.020	0.056	0.028	0.008	0.51	0.57
C24:0	0.078	0.13	0.11	0.096	0.02	0.76	0.40
C24:1 *cis*-15	0.045	0.040	0.028	0.059	0.008	0.74	0.32

^1^ CON, LCaP, MCaP, and HCaP were mixed with 0, 200, 350, and 500 g/d calcium propionate per cow, respectively. ^2^ SEM: Standard error of the mean. ^3^ L: Linear effects of calcium propionate supplementation; Q: Quadratic effects of calcium propionate supplementation. The terms *cis* and *trans* refer to the molecular configuration of fatty acids.

**Table 6 animals-15-02995-t006:** Effects of dietary supplementation with different levels of calcium propionate on the proportions (% of total fatty acids) of milk fatty acids according to the carbon chain length and degree of unsaturation of dairy cows during early lactation (1–35 DIM).

Class of Fatty Acids ^1^	Treatments ^2^				SEM ^3^	*p* Value ^4^	
	CON	LCaP	MCaP	HCaP		L	Q
SFA	66.41	64.10	65.34	63.52	0.83	0.32	0.88
MUFA	29.97	32.71	31.70	33.11	0.83	0.26	0.70
PUFA	3.62	3.19	2.96	3.37	0.10	0.11	0.01
UFA	33.59	35.90	34.66	36.48	0.83	0.32	0.88
SCFA	6.61	5.84	5.98	5.64	0.21	0.14	0.63
MCFA	48.27	47.75	49.66	46.57	0.82	0.65	0.41
LCFA	45.12	46.41	44.37	47.79	0.92	0.46	0.57
Trans	0.73	1.07	1.17	0.88	0.08	0.39	0.07
SFA/UFA	2.22	1.95	2.23	1.91	0.12	0.55	0.95

^1^ SFA: Saturated fatty acids; MUFA: Monounsaturated fatty acids; PUFA: Polyunsaturated fatty acids; UFA: Unsaturated fatty acids; SCFA: Short-chain fatty acids (acyl chain < 11 carbon atoms); MCFA: Medium-chain fatty acids (acyl chain > 10 carbon atoms and < 17 carbon atoms); LCFA: Long-chain fatty acids (acyl chain > 16 carbon atoms); Trans: Total trans fatty acids. ^2^ CON, LCaP, MCaP, and HCaP had 0, 200, 350, and 500 g/d calcium propionate per cow, respectively. ^3^ SEM: Standard error of the mean. ^4^ L: Linear effects of calcium propionate supplementation; Q: Quadratic effects of calcium propionate supplementation.

## Data Availability

Data are available from the corresponding author upon request.

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
