# Peer review of "Optimal Calcium Propionate Supplementation in Early-Lactation Dairy Cows Improves Milk Yield and Alters Milk Composition"

_animals, 2025, doi:10.3390/ani15202995_

Round 1
Reviewer 1 Report
Comments and Suggestions for Authors
This study evaluated the effect of calcium propionate supplementation levels in post-partum dairy cows on productive performance, milk mineral composition, and fatty acid profiles. This study supplemented 3 different levels of calcium propionate to the cows from calving to 35 days in milk and measured several outcomes to measure productivity.
In general, this is a well-designed experiment with sufficient power to measure the goal of the study. I only have a few minor comments.
General comment
The title is too long; consider shortening it. You could also just use the main finding from the study as the title.
I believe the maximum word count allowed for the abstract for this journal is 200. Your abstract seems a bit too long; consider shortening it and making it more concise.
The keywords should be something that is not in the title, considering using other keywords for better indexing purposes.
For Table 3, I suggest only running L and Q OR treatment, time, and treatment*time. When you run both and put them in the table, it is a bit redundant. As you tested whether more CaPro will be better with the linear and quadratic test, treatment basically represents the same thing; it’s just one is continuous and one is factor. Pick one and remove another statistical analysis. Also, since you are not talking about the time effect on the measured parameters.
Same thing with Tables 4 and 5, suggest removing the treatment, time, and treatment*time p-value. Since you have Figures 1 and 2 to explain the interaction, suggest making your result more concise.
Consider making your Table 6 more concise.
For your discussion, make sure you clearly separate the findings from the previous study using citations and findings from your study using words like in the current study, from our experiment, etc. It gets confusing when you are making comparisons without proper distinction.
Specific comment:
L10: Define NEB at first use
L68: citation format
L68-72: Suggest elaborating on those previous studies so that it could explain the novelty of your study in comparison to those. For example, you supplemented during early lactation, when did the previous studies supplement theirs? Are there differences in the supplementation levels used? etc.
Table 1. Just out of curiosity, will it be helpful if you provide the DCAD values of the treatment diets to show the reader the differences in DCAD between treatments?
L 154: citation format
Table 3: There is a tendency for Lactose that you didn’t mention. Also add units to all items
L234: indicate the direction of the alteration
L333-334: Which calcium propionate groups?
L337-338: Is it statistically significant?
L341-344: I don’t recall seeing blood Ca data. Where was it? What is your blood calcium level in comparison to the previous studies? Similar? Higher?
L393: indicate the direction of change
Reviewer 2 Report
Comments and Suggestions for Authors
Reviewer Comments on Manuscript: "Effects of calcium propionate supplementation levels in post-partum dairy cows on productive performance, milk mineral composition, and fatty acid profiles"
This manuscript presents a well-designed and interesting study on a relevant nutritional intervention for postpartum dairy cows. The topic is significant for improving dairy cow health and productivity. The experimental design is reasonable, the methods are generally robust, and the results are valuable. However, before considering publication, there are several points that need to be clarified and elaborated to enhance the impact and scientific rigor of the manuscript.
- Abstract: L30-31, Suggesting include the key numerical values of milk yield for the main finding in the abstract replace “with the highest value observed in the MCaP group” to “with the highest value (XX kg/d) observed in the MCaP group.
- L18 “…for mitigating hypocalcemia and ketosis…” chang to “…for preventing hypocalcaemia and ketosis…”
- L 39: “…concurrently improving lactation performance” change to “…while improving lactation performance.”
- Introduction: L 67: “gluconeogenetic precursors” change to “gluconeogenic precursors”.
- L78-83: The hypothesis is clear but could be more concise and impactful. Consider refining the hypothesis statement for greater clarity. Suggestion: “We hypothesized that dietary calcium propionate supplementation would enhance subsequent lactation performance by improving mineral homeostasis and altering fatty acid metabolism in early lactation, reflecting a reduced reliance on body tissue mobilization."
- L96-87 “The cows in the experiment were loosely housed in individual stalls” change to “The cows were housed in individual stalls”.
- Result: L 239, L245, The results mention significant treatment × time interactions for several variables (milk K, Fe, fatty acids). The descriptions in the text are somewhat difficult to follow. Ensure the figure captions are extremely detailed, explaining the patterns visible in the graphs. The text should guide the reader to the figures (e.g., "As shown in Figure 1, the milk K concentration in the HCaP group was significantly lower at 21 DIM...").
- Discussion: L 334, deleted “The calcium propionate-supplemented groups presented greater milk yield.” (said twice).”
- Terminology must be standardized: use either “early-lactation” or “early lactation” consistently; “gluconeogenetic” should be “gluconeogenic”.
Reviewer 3 Report
Comments and Suggestions for Authors
Review of manuscript; Effects of calcium propionate supplementation levels in post- partum dairy cows on productive performance, milk mineral composition, and fatty acid profiles
- The abstract needs to be changed: please add the purpose of the research and the conclusion resulting from the research.
- The introduction needs to be rewritten. A brief note should be added explaining the physiological causes of ketosis and hypocalcemia. In which cow husbandry system does this disease occur? (extensive, intensive system)
- The material and research methods should be supplemented. Were the cows matched in terms of age (which was calving), how were they kept before the start of the experiment?. Was the calcium supplement (5%, 10%) dictated by the literature data, or did the authors choose the doses themselves? . Were the milk samples taken individually from each cow, or were they pooled samples for the group?. Why were the samples taken in the morning, at noon, and in the evening mixed (4:3:3), and why was the fatty acid content not determined separately for each sample?. The methodology should include when calves were separated from their mothers. When were they weaned, at what time of year?
- The results are described correctly, and the tables accurately present the results.
- The summary needs to be changed; it is a description of the information discussed. You should write down what the research revealed that is important.
Round 2
Reviewer 3 Report
Comments and Suggestions for Authors
I accept the manuscript, the additions are fine.